# 3D-Printed Tumor-on-a-Chip Model for Investigating the Effect of Matrix Stiffness on Glioblastoma Tumor Invasion

**DOI:** 10.3390/biomimetics8050421

**Published:** 2023-09-11

**Authors:** Meitham Amereh, Amir Seyfoori, Briana Dallinger, Mostafa Azimzadeh, Evan Stefanek, Mohsen Akbari

**Affiliations:** 1Department of Mechanical Engineering, University of Victoria, Victoria, BC V8P 5C2, Canada; mamereh@uvic.ca (M.A.); amirseyfoori@uvic.ca (A.S.); mazimzadeh@uvic.ca (M.A.); 2Laboratory for Innovations in MicroEngineering (LiME), Department of Mechanical Engineering, University of Victoria, Victoria, BC V8P 5C2, Canada; bdalli@live.ca (B.D.); evanstef@uvic.ca (E.S.); 3Centre for Advanced Materials and Related Technologies (CAMTEC), University of Victoria, Victoria, BC V8W 2Y2, Canada; 4Terasaki Institute for Biomedical Innovations, Los Angeles, CA 91367, USA

**Keywords:** 3D-printing, tumor-on-a-chip, glioblastoma, in silico model

## Abstract

Glioblastoma multiform (GBM) tumor progression has been recognized to be correlated with extracellular matrix (ECM) stiffness. Dynamic variation of tumor ECM is primarily regulated by a family of enzymes which induce remodeling and degradation. In this paper, we investigated the effect of matrix stiffness on the invasion pattern of human glioblastoma tumoroids. A 3D-printed tumor-on-a-chip platform was utilized to culture human glioblastoma tumoroids with the capability of evaluating the effect of stiffness on tumor progression. To induce variations in the stiffness of the collagen matrix, different concentrations of collagenase were added, thereby creating an inhomogeneous collagen concentration. To better understand the mechanisms involved in GBM invasion, an in silico hybrid mathematical model was used to predict the evolution of a tumor in an inhomogeneous environment, providing the ability to study multiple dynamic interacting variables. The model consists of a continuum reaction–diffusion model for the growth of tumoroids and a discrete model to capture the migration of single cells into the surrounding tissue. Results revealed that tumoroids exhibit two distinct patterns of invasion in response to the concentration of collagenase, namely ring-type and finger-type patterns. Moreover, higher concentrations of collagenase resulted in greater invasion lengths, confirming the strong dependency of tumor behavior on the stiffness of the surrounding matrix. The agreement between the experimental results and the model’s predictions demonstrates the advantages of this approach in investigating the impact of various extracellular matrix characteristics on tumor growth and invasion.

## 1. Introduction

Glioblastoma, an aggressive form of primary brain cancer with a low survival rate and limited treatment options, is one of the deadliest types of cancer. Despite advances in treatment, patients typically survive less than 15 months after diagnosis. This is because it is difficult to surgically remove the tumor due to its rapid growth and invasion into adjacent regions of the brain [1,2]. Understanding tumor invasion mechanisms may reveal therapeutic targets to stop or slow tumor progression, improving patient outcomes and leading to improved treatments. There are four different routes or patterns that the invasion of GBM can take: through the leptomeningeal space, the brain parenchyma, the white matter tracts, or the perivascular space. Depending on the invasion route, the GBM cells can produce varying degrees of heterogeneity when they invade other parts of the brain. This can make it more difficult to determine the prognosis and appropriate treatments for GBM [3,4].

Invasion is one of the key hallmarks of tumors [5]. Gaining insights into tumor invasion mechanisms facilitates the development of combination therapies designed to disrupt key invasion pathways in the primary tumor [6]. In addition, invasion mechanisms and patterns may be specific to cancer type. Therefore, understanding tumor invasion may help develop a personalized approach which not only targets the cancer’s unique characteristics but also enables early detection through diagnostic tools that are capable of identifying invasive behavior [7]. Moreover, targeted therapy can benefit from knowledge of tumor invasion, as these therapies can then be optimized using knowledge of how tumors invade surrounding tissues, ensuring that drug delivery systems effectively reach and target invasive cells, including those resistant to conventional treatments [8]. Overall, understanding the mechanism of cancer invasion enhances comprehensive strategies and treatment efficacy by addressing multiple aspects of cancer growth and spread, simultaneously.

Tumor modeling has revealed that the invasion and formation of invasive structures in glioblastoma cells varies depending on the stiffness of their matrix [9]. Recent research has demonstrated a direct connection between the ECM of the brain and its modifications surrounding the GBM niche [10]. In the initial phases of GBM, local changes in the ECM occur throughout the brain [11]. As the disease progresses, these changes also facilitate the invasive behavior of the tumor. The variations observed in the ECM of GBM primarily involve modifications in the expression levels of multiple components. They can influence rapid biological and physiological shifts within cellular functions and give rise to the development and progression of tumors [12]. Therefore, stiffness of the ECM represents a key mechanical factor that influences the progression of GBM.

Understanding this mechanism of invasion may help treat invasive tumors cells and improve patient prognoses [13,14]. Recently, there has been an increasing focus on creating three-dimensional cancer models using spheroids, which involves forming clusters of tumor cells with a spherical shape. Spheroids recapitulate cell–cell interactions, ECM remodeling, and intercellular signaling [15,16], which are important components of the complex glioblastoma tumor microenvironment. Creating tumoroids can be done using novel approaches, such as 3D bioprinting with different cells, extracellular matrices, and blood vessel networks [17,18]. Refs. [19,20] show advantages over earlier methods, such as hanging drops [21] and microwells [22]. There are many different 3D bioprinting methods, and digital light processing (DLP) 3D bioprinting can be a high resolution, fast, and high-throughput choice producing highly vascularized microstructures similar to those found in native tumors using cells and photocurable biomaterials [23,24,25].

Mathematical models are transforming our comprehension of biological systems through in silico modeling. This enables us to understand the interactions that underlie complex behavior in invasive tumors [26,27]. These models can be generally categorized into either continuum or discrete individual-based models [28,29]. Over the past several years, a third category that has the advantages of continuum and discrete models, the so-called Hybrid Discrete-Continuum (HDC) model, has been developed [20]. HDC models use continuum fields to build up the discrete approach; therefore, the process is at both micro and macro scales. The continuum part provides information on the field variables, such as densities and velocities, which are inputted into a discrete part to update the cellular interactions. In addition, tumor heterogeneity and TME components are key factors in developing realistic in silico models [30].

In this study, we investigated how the stiffness of the matrix influences the invasive behavior of human glioblastoma tumoroids. An open surface tumor-on-a-chip platform was 3D printed using a photocurable poly-(ethylene glycol) diacrylate (PEGDA) hydrogel. The chip models the human glioblastoma tumoroid microenvironment in separate chambers with embedded collagen. The current tumor-on-a-chip platform consists of microchannels that flow around a central chamber. This chamber is where tumoroids are embedded in the ECM. The microchannel around the ECM chamber is multifunctional and can either deliver localized therapy or generate a vascularized lumen network that surrounds the microtumor tissue. Additionally, this platform can conduct four experimental conditions at once in a single well, which is compatible with common well-plate tissue cultures. The platform creates a symmetrical distribution of MMP1 through the ECM, resulting in a gradient of ECM stiffness for invasion analysis. This even distribution allows for the study of the effect of stiffness gradient on tumoroid invasion patterns, such as finger and ring types, from all directions. The microfluidic platform created a gradient of extracellular matrix stiffness within the collagen matrix. The impact of this gradient on the tumoroid’s invasion behavior under varying gradients was investigated. To better interpret the experimental data, a hybrid mathematical model based on a continuous reaction–diffusion model was used to model tumor growth. In contrast, a discrete model was used to model cell movement into the surrounding tissue. This study uses experimental and computational methods to show how the matrix’s stiffness affects the glioblastoma tumoroid’s invasion pattern.

## 2. Materials and Methods

### 2.1. Materials

PEGDA (Mw 700 Da), LAP, and tartrazine were purchased from Sigma-Aldrich (St. Louis, MO, USA). Fetal bovine serum (FBS), phosphate-buffered saline (PBS), Dulbecco’s modified Eagle’s medium (DMEM) and penicillin were purchased from Gibco (Grand Island, NE, USA). The 10× phosphate buffered saline (PBS), 0.5 N NaOH and bovine collagen type 1 (10 mg/mL) were purchased from Advanced BioMatrix Inc. (San Diego, CA, USA). Collagenase type 2 was purchased from Worthington Biochemical Corporation (Lakewood, NJ, USA). Tumoroids culture plate was purchased from Apricell Biotechnology Inc. (Victoria, BC, Canada). The rest of the suppliers are indicated throughout the text.

### 2.2. DLP Stereolithography Apparatus

The bioprinting apparatus used was a DLP stereolithography printer (Lumen X, Cellink, San Carlos, CA, US). The main hardware components of the printing apparatus were the DLP projector, and the *z*-axis stage and motor. The projector power output can be varied from 10–30 mW/cm^2^ and the wavelength of the projected light was 405 nm (visible light). The XY pixel resolution of the projector was 50 μm, the Z resolution of the motor was 5 μm, and the minimum printable layer height was 50 μm. The total build volume is 64 × 40 × 50 mm. Additional components of the device are the build platform and vat. The hydrogels are built in a layer-by-layer fashion on the build platform, which attaches to the Z stage. The vat that contains the prepolymer solution is a petri dish coated with a layer of poly(dimethyl-siloxane) (PDMS). The PDMS prevented the construct from sticking to the vat during the crosslinking process.

### 2.3. 3D Printing PEGDA Hydrogel

The prepolymer solution is composed of 15% *v*/*v* PEGDA (700 Da, Sigma Aldrich), 0.6% *w*/*v* lithium phenyl-2,4,6-trimethylbenzoylphosphinate (LAP; Sigma Aldrich), and 2.5 mM tartrazine (Sigma Aldrich) in distilled water. The components are measured into a falcon tube and dissolved by heating to 65 °C and by vortexing. The design that is to be printed is made in SolidWorks and converted to an STL file. To print, first the surface of the build platform is sonicated in ethanol for three minutes and set aside to dry. Next, the PDMS vat is cleaned with isopropanol and distilled water. With the build platform and vat in place, the printer Z stage is calibrated. The exposure time per layer is five seconds per layer and the power intensity of the projector is 21.5 mW. The first layer time scale value is lesser (3×) for smaller constructs and higher (5×) for larger constructs. The added exposure time given to the first layer of the print helps the construct adhere to the build platform. After the print is complete, the build platform is removed from the Z stage and the construct is carefully removed using a plastic razor. The construct is placed in a beaker containing distilled water, and this water is exchanged periodically to remove excess tartrazine from the construct.

### 2.4. Optimization of PEGDA Photoink

The prepolymer PEGDA solution was optimized for printing small horizontal channels. A systematic set of experiments were conducted to optimize each component of the prepolymer solution individually. The parameters of interest were LAP concentration, tartrazine concentration, and projector power, while the PEGDA concentration was maintained at 15%. One at a time, each component was varied (three values) while the other components were kept constant, and the best performing formulation was maintained for the next component test. Ideal exposure time per layer varied widely depending on the components in the prepolymer solution, so set values could not be predetermined prior to testing and a few values were tested with each formulation. Performance of the printed hydrogel was based on material integrity and the smallest printable channel diameter. A horizontally oriented channel model with varying channel sizes (1.0, 0.50, 0.40, 0.30, 0.20, and 0.10 mm) was printed with each hydrogel formulation to test its performance. The values tested were 1.02, 2.04, and 3.06 mM LAP, 1.0, 2.5, and 4.0 mM tartrazine, and 14.0, 21.5, and 25.0 mW/cm^2^ projector power. The optimized formulation consisted of 15% PEGDA, 2.04 mM LAP, and 2.5 mM tartrazine printed with projector power of 21.5 mW/cm^2^ and 5 s per layer exposure time.

### 2.5. Glioblastoma Cell Culture and Tumoroid Preparation

Human glioblastoma (hGB) U251cell lines expressing mKate, a red fluorescent protein (developed by Dr. Marcel Bally, Experimental Therapeutics, British Columbia Cancer Agency, Vancouver, BC, Canada) were cultured in DMEM supplemented with 10% FBS, 100 IU/mL penicillin and 100 mg/mL streptomycin. The U251 cells were incubated at 37 °C in a humidified atmosphere of 5% CO_2_, and the culture medium was replaced every other day. At 90% confluency, cells were trypsinized into a single cell suspension for 5 min and, to neutralize trypsin, 6 mL of growth medium was added. The cell suspension was centrifuged at 1000 rpm for 5 min to avoid dead cell sedimentation. After removing the supernatant, cells were suspended in 1 mL of medium and counted using a standard hemocytometer. U251 multicellular tumoroid were grown in EZ-seed 3D culture plate from Apricell Biotechnology Inc. In the 3D tumoroid culture plate, U-shaped microwells made of agarose hydrogel served as self-filing microwell arrays (SFMA). To generate tumoroids, agarose microwells were firstly submerged into 1.2 mL culture media per well in 12 well plates, followed by cell seeding with a concentration of 500 × 10^3^ cells/200 μL. Cell-loaded microwells were incubated for 4 days to form compact and round tumoroids.

### 2.6. 3D Tumoroids Culture On-a-Chip

The four-day old cultured U251 tumoroids were collected from the SFMA platform by gentle washing with culture media. To mimic the tumor ECM condition of glioblastoma, bovine fibril collagen with a concertation of 4 mg/mL was used as the main hydrogel component of the model. To prepare the hydrogel/tumoroids suspension, first the pH and ionic concentrations of the collagen solution were adjusted by adding 10× PBS and 0.5 N NaOH to the stock solution of collagen with a ratio of 1:1:8. Afterwards, a certain volume of culture media containing one tumoroid was immediately mixed with collagen solution to achieve the desired concertation. The final collagen solution was gently introduced into the central chamber of the 3D printed microfluidic device and then incubated in an incubator at 37 °C for 45 min to complete crosslinking.

### 2.7. Enzyme Treatment and Cell Invasion Analysis

Effect of collagenase on the invasion length of the tumoroids inside the ECM hydrogel was studied using two different concentrations of collagenase type I, 0.001 and 0.01 mg/mL. Collagenase solutions with each concentration were made in culture media and introduced symmetrically through the inlet of the microfluidic platform and filled the side chamber of the device. For the invasion studies, the treatment was started immediately after tumoroid seeding and enzyme treatment lasted over three days. The effect of collagenase with different concentrations on tumoroid invasion was analyzed and quantified every day by fluorescent imaging using the Zeiss microscope.

### 2.8. Mechanical Properties of the Collagen Hydrogel

Rheological properties of the collagen hydrogel with and without treatment with collagenase was studied using an Anton Paar (MCR 302 Rheometer, Graz, Austria) rheometer with 25 mm diameter sandblasted plate-to-plate geometry (PP25/s) with 1 mm gap. An amplitude sweep test at the frequency of 1 Hz was performed and the linear viscoelastic region was determined for all gel formulations. Variations in shear stress as well as the viscoelastic parameters such as storage modulus, loss modulus, and complex modulus as a function of shear strain were recorded over the range of 0.1–10%. Oscillatory frequency sweep measurement was carried out at a 0.1–100 rad·s^−1^ in the viscoelastic region (0.5%) for all conditions to further characterize viscoelastic properties. Variations of the storage modulus (*G*′), loss modulus (*G*″), complex modulus (*G**), and complex viscosity (*η**) against angular frequency sweep were also recorded.

### 2.9. Statistical Analysis

All experiments were repeated three times and the average and the standard deviation were reported. Significance analysis was performed using two-way ANOVA analysis. Differences were considered statistically significant at *p*-value < 0.05.

## 3. Results and Discussion

### 3.1. PEGDA Bioprinting Parameters Optimization

The Lumen X DLP printer was used to print the tumor-on-a-chip platform, Figure 1a. To optimize channel printing, horizontal and vertical channels were printed for ranges of variables. The prepolymer PEGDA solution was optimized to print small, horizontally aligned channels. The variables to be optimized were the LAP concentration, tartrazine concentration, projector power, and exposure time per layer (Figure 1b). Each variable was tested sequentially with three different test values while all other components were kept constant except for exposure time per layer which had to be adjusted for each test case.

PEGDA concentration was not a variable that was tested and was maintained at 15% during the optimization process. For each variable, a horizontal channel model with varying channel sizes from 1.0 mm to 0.1 mm in diameter was printed with each PEGDA formulation. The smallest channel that could be printed was recorded from each test, with the goal being to print the smallest possible channels (Figure 1c). LAP concentration was the first variable to be tested, once the best LAP concentration was determined, it was carried forward and then tartrazine concentration was tested, and once the best tartrazine concentration was determined, it was carried forward to the projector power testing. At the end of this optimization process, the PEGDA formulation which produced the smallest channel diameter was 15% PEGDA, 2.04 mM LAP, 2.5 mM tartrazine, 5 s/layer exposure, and 21.5 mW/cm^2^ power intensity. This optimal formulation produced hollow horizontal channels as small as 0.3 mm in some tests, and consistently produced hollow 0.4 mm diameter channels. The relationship between LAP and tartrazine concentration in the PEGDA formulation is shown in the figure. When the tartrazine concentration was too low, 1 mM tartrazine with 1.02 mM and 2.04 mM LAP, none of the channels printed hollow. LAP concentration did not have much effect on channel printing, but overall made the printed gel stiffer at higher concentrations. To test the accuracy of printing, a microfluidic chip with ten channel diameters from 1.0 to 0.1 mm was 3D printed, the diameter of each channel was measured using ImageJ, and the measured values were compared to the nominal diameters, i.e., horizontal lines (Figure 1d).

### 3.2. Tumor-on-a-Chip Platform

Directional migration of the GBM is a biological phenomenon dependent on multiple factors including chemotaxis and hapto-taxis [31,32,33]. Collagenase type I (MMP1) as a member of the MMP family secreted in the tumor microenvironment has a massive effect on changes to local mechanical properties of the tumor ECM and subsequent migration of the tumor cells [34]. In this regard, we aimed to measure the effect of collagenase on the invasion behavior of U251 glioma tumoroids inside the fibril collagen hydrogel which mimics the main component of the ECM of the brain tumor microenvironment.

Using a tumor-on-a-chip platform with multiple chambers, we cultured U251 tumoroids in collagen, with the capability of separately inducing each chamber, with different concentrations of collagenase (Figure 2a). This platform can conduct four experimental conditions at once in a single well, which is compatible with common well-plate tissue cultures. The microfluidic chip has a perfused round microchannel surrounding the central chamber. Tumoroids will be embedded within ECM inside the chamber, and they are perfused with the media and collagenase included media through the surrounding channel. This design creates a symmetrical distribution of MMP1 through the ECM, resulting in a gradient of ECM stiffness for invasion analysis. Each chamber contains an inlet and outlet channel, and collagenase solution with different concentrations was added to the microchannels from the inlet of the device. The microchannel around the ECM chamber is multifunctional and can potentially either deliver localized therapy or generate a vascularized lumen network that surrounds the microtumor tissue. A self-filling microwell array (SFMA) from Apricell Biotechnology Inc. was utilized to generate uniform tumoroids, which were then mixed with collagen and placed inside chambers (Figure 2b,c) [35]. The thin floor of the chambers enables clear microscopy imaging of growth and invasion of tumoroids over time (Figure 2d). Collagenase type I solutions with concentrations of 0.01 and 0.001 mg/mL were introduced to the collagen embedded tumoroids in the model through the peripheral channels of the microfluidic platform and incubated for 48 h. This platform enables the quantitative analysis of the tumoroid invasion over time. Our developed 3D printed microfluidic platform models how solid tumors behave in their extracellular matrix. Our platform can potentially employ various strategies that consider gradient physio–mechanical properties in cancer studies. The circular microchannels surrounding the central ECM chamber are designed to serve as a localized depot for chemo attractants. This enables us to model how tumor cells invade outward from the core tissue. Our platform can be used as an in vitro model to evaluate therapeutic payloads that trap invaded tumor cells. This leads to more efficient cancer therapy for invasive solid tumors.

### 3.3. Mathematical Modeling

Cancer cell migration, which plays a pivotal role in tumor invasion, encompasses diverse cellular mechanisms [36,37,38]. Multiple modes of cell migration, including individual and collective motions, contribute to the overall patterns of invasion. These types of migration can potentially give rise to finger-type and ring-type invasion patterns, respectively [39,40]. The former accounts for the migration of single cells into the surrounding tissue and the latter corresponds to the collective motion of cells in the form of a ring. In single cell motion, cells gain the ability to move by adopting either a mesenchymal or amoeboid morphology [41]. However, in the case of collective cell migration, cells retain their connections with neighboring cells by maintaining intercellular adhesion and form a leading edge that extends into the surrounding tissue [42].

In this section, the evolution of the tumor is mathematically modeled using an HDC technique which utilizes a continuum model for the derivation of a discrete model [39]. The cell flux, diffusion of nutrients and collagenase, as well as the degradation of collagen, are predicted using a continuum model. The generation of cell flux primarily occurs through the interplay of diffusive and adhesive forces, as well as the directional movement resulting from chemotaxis. We adopted a nonconvex form of free energy and a standard chemotactic model presented in [40] to model the collective motion of cells (Figure 3a). Additionally, the motion of single invasive cells, including random and directional motions, was predicted by a discrete model (Figure 3b). A reaction–diffusion model that consists of a system of partial differential equations (PDEs) represented the collective motion of cells. In this model, state variables, such as concentrations of cells, nutrients, collagen, and collagenase, were assumed to be continuous, and their local changes are obtained by the following governing equations [39].
(1)$$\frac{\partial C_{p}}{\partial t}=\nabla \;\cdot \;\left(J_{f}+J_{d}\right)+C_{p}\;\eta_{p},$$
(2)$$\frac{\partial n}{\partial t}=D_{n}\nabla^{2}n-C_{p}\;\lambda_{p},$$
(3)$$\frac{\partial M}{\partial t}=D_{M}\nabla^{2}M-\zeta Mf,$$
(4)$$\frac{\partial f}{\partial t}=-\delta Mf,$$
where t is time, and $C_{p}$, n, f, M are concentrations of proliferative cells, nutrient, collagenase and collagen, respectively. Not that potential inhomogeneity in cells population within the tumoroids was neglected. This assumption, which simplifies the model, becomes more precise as the size of tumoroids becomes smaller, since they do not develop hypoxia. We should also note that all components in the experimental part, including tumoroids, media, collagen, and collagenase, are continuum materials, which is aligned with the assumption in the continuum model. $J_{f}$ and $J_{d}$ are the free energy and chemotaxis fluxes. $D_{n},D_{M}$ are diffusivity coefficients of nutrient and collagenase, and ${C_{p}\eta }_{p}$ and ${C_{p}\lambda }_{p}$ are rates of proliferation and nutrient consumption, respectively. Following [39], we assumed that the rate of collagen degradation is linearly proportional to the concentration of collagenase and, likewise, collagenase degradation is proportional to the collagen concentration, where ζ and δ denote the linear proportionalities. Denoting the tumor domain by Ω and the free boundary by δΩ, Equations (1)–(4) are subjected to the following boundary and initial conditions:(5)$$BCs:C_{p}\left(x,t\right){|}_{\delta \Omega }=C_{0},\;n\left(x,t\right){|}_{\delta \Omega }=n_{0},\;M\left(x,t\right){|}_{\delta \Omega }=M_{0},$$
(6)$$ICs:C_{p}\left(x,t=0\right)=C_{i},\;n\left(x,t=0\right)=n_{i},\;M\left(x,t=0\right)=0,\;f\left(x,t=0\right)=f_{0},$$
where x is the position, {$C_{0},\;n_{0}$, $M_{0}\}$ are the boundary conditions of cells, nutrient, and collagenase, and ${\{C}_{i},\;n_{i}$, $M_{i}\}$ are the initial concentrations of cells, nutrient and collagen, respectively. Here, we only considered the axisymmetric solutions due to the symmetric boundary and initial conditions. The governing equations of the continuum model predict the ring-type invasion of tumor and the distribution of nutrients, collagen, and collagenase densities. The next section presents the discrete technique which employs the state variable fields obtained from the continuum model to derive cellular interactions such as single cell migration.

Cellular migration mechanisms depend on morphology and external stimuli. Invasive tumor cells become morphologically polarized and develop membrane protrusions, allowing them to reach forward [43]. This motion is called mesenchymal when cells move via traction and adhesion to the ECM. On the other hand, the motion can be amoeboid when cells squeeze through the pores in the ECM, which is relatively fast and does not require strong adhesion forces [44]. The former type of migration is directional (hapto-taxis) and the latter can be both directional and random (chemo-kinesis) [45,46] (Figure 3a). To capture both mechanisms, we write the cell flux $J_{i}$ as a combination of random and directional fluxes
(7)$$J_{i}=-D_{i}\nabla C_{i}+\chi_{hap}C_{i}\nabla f,$$
where $D_{i}$ and $\chi_{hap}$ are coefficients of random and haptotactic migrations. Polar coordinates, $x=re_{r}\;\left(\theta \right),$ are used as they well suited to represent the shape of tumoroids. Considering temporal (i), radial (j) and angular (k) discretizations, finite difference approximations transform the differential equations into an algebraic system of equations where $C_{j,k}^{i}$ is the discretized cell concentration [39].

Equation (8) is used to find the probabilities of cell motions in different directions on the discretized plane. Implementing central finite differences technique and solving for $C_{j,k}^{i+1}$ gives
(8)$$C_{j,k}^{i+1}=P_{0}\;C_{j,k}^{i}+P_{1}\;C_{j+1,k}^{i}+P_{2}\;C_{j-1,k}^{i}+P_{3}\;C_{j,k+1}^{i}+P_{4}\;C_{j,k-1}^{i}$$
where $P_{0},P_{1},P_{2},P_{3}$ and $P_{4}$ are proportional to probabilities of the cell staying stationary, or moving in directions of +r (forward), −r (backward), +θ counterclockwise and −θ clockwise, respectively [39]. These movement probabilities are functions of the cell’s random motion and the collagen gradient and link the discrete model to the continuum model. Therefore, the discrete model governs the migration of individual cells based on their interactions with the ECM.

The complex dynamics of the system of the discrete model, which captures the cellular processes, is coupled with the continuum model. The simulation flowchart depicts the relationship between continuum variables and cellular processes, such as proliferation, cellular age, mechanical stress, etc. Using this model, the system receives updated information from the continuum model and applies this in the discreet model (Figure 3c). Model parameters are presented in Table 1.

$\eta_{p}$$\lambda_{p}$ζAs shown in Figure 4a, there is a gradual and time-dependent increase in invasion length of the tumoroids with no collagenase treatment. It is also demonstrated that the invasion area after 72 h is significantly increased in both collagenase treatment conditions compared to the non-treated sample. Quantification of the invasion area did not show any significant difference between the treated and non-treated conditions at day 0, while the invasion length of the tumoroids treated with 0.01 mg/mL collagenase is significantly increased in comparison with less concentration of collagenase (0.001 mg/mL) at day 1 (Figure 4b). The difference in invasion potential of treated and non-treated samples remained consistent at day 3 while there was no statistical significance between the invasion length of the tumoroids with 0.01 and 0.001 mg/mL collagenase. In the current model, we observed that tumoroids embedded with collagen and treated with varying concentrations of collagenase showed both finger-type and ring-type invasion patterns, previously observed in glioblastoma tumors [39,53]. It was reported that collagen enhances collective migration in glioblastoma cells, whereby a cluster of cells can invade into the surrounding gel [50]. Moreover, finger-type migration in glioblastoma tumoroids was shown to be the main invasion pattern [36]. Here, we observed the same invasion patterns; (i) cluster migration of cells which manifests a ring, due to the symmetry of model, moving away from the tumoroid, and (ii) individual cells migrating from the ring and forming a finger-type pattern. We conducted quantitative analyses of these two patterns in Figure 4c and found that finger-type invasion was more dominant than ring-type invasion in our in vitro model (Figure 4c). Moreover, by increasing the concentration of collagenase treatment, tumor cells invaded under the finger-type pattern showed a faster rate than the cells invaded under the ring-type pattern.

The process of invasion involves the local degradation of collagen molecules in the ECM by MMP1 secretion, increasing the porosity of the ECM hydrogel network [54]. To investigate the effect of MMP1 on the invasion pattern of tumoroids, matrix stiffness was used as an indication of changes in the mechanical properties of the ECM after the secretion of collagenase from tumor cells or other elements in the tumor microenvironment during invasion. Our aim was to demonstrate how changing the ECM stiffness in an in-vitro model can complementarily explain our findings, which have been added to the manuscript. To quantify the effect of collagenase on tumoroid ECM, we measured the variation of viscoelastic properties of collagen. Storage modulus (elastic modulus) and loss modulus are representative mechanical properties of viscoelastic materials, such as hydrogels [55]. Storage modulus (*G*′) represents a material’s ability to store and recover elastic energy when subjected to an applied force or deformation, and loss modulus (*G*″) represents the viscous or damping behavior of a material. These two properties together provide insights into the mechanical behavior of a material across a range of deformation. The interplay between them defines how the material responds to different mechanical forces and environments. Therefore, they are crucial for designing biomaterials.

Invasion behavior of tumoroids was aligned with the mechanical properties trend of the collagen hydrogel in the presence of different collagenase concentrations (Figure 5). Higher concentration of collagenase significantly reduced both storage and loss moduli. According to previous results in the literature, upregulation of matrix metalloproteinase 9 (MMP-9) and enhanced tumor cell invasion are the downstream consequence of TGF-*β*1 secretion through autocrine signaling from glioma cells [56]. The stiffness of the extracellular matrix (ECM) in solid tumors is a critical physical factor that affects their ability to invade healthy tissue. Observations in vivo show that the ECM’s stiffness changes during carcinogenesis stemming from ECM remodeling, resulting from the diverse compositions and densities of collagen [54]. Integrin receptors play a crucial role in transmitting physical cues from the ECM’s stiffness to tumor cells, prompting mechano-transduction events that modify their invasive potential. Studies indicate that changes in tumor tissue matrix stiffness substantially impact how tumor cells respond to chemotherapy and immunotherapy [57]. In vitro model findings are consistent with in vivo observations of the tumor microenvironment’s mechanical behavior during progression and invasion within the ECM. These results could be valuable for cancer biologists and oncologists seeking to understand the significant impact of ECM stiffness on tumor behavior. Hereby, our in vitro model can be used to predict the migration behavior of the glioma tumor cells in a 3D microenvironment in response to localized TME mechanical stiffness alterations. This model can provide an opportunity for studying the effect of TME associated signaling pathways on the function of the tumor cells in ECM and their crosstalk with the other components of the tumor microenvironment. Moreover, this 3D model opens a new venue to study our hypothesis on directing the migration of the tumor cells from the deeper area of the white matter within the brain tissue toward the external source of the anticancer drug. We tested this hypothesis by using MMP 1 with the aim of making a gradient of the collagenase concentration within the hydrogel, leading to partial degradation of the collagen ECM. This model could facilitate measurement of the tumor cells’ invasion from the stiffer areas to the less stiff zone of ECM.

Similar to other mathematical techniques, hybrid discrete-continuum (HDC) models come with certain limitations [58]. For instance, they often demand sophisticated numerical techniques to integrate the discrete and continuous components, leading to increased computational complexity. They sometimes require a large number of model parameter estimations, which can be difficult, and validating the model against both discrete and continuous experimental data is complex. Additionally, bridging different scales and granularities while considering data requirements and assumptions can present significant technical hurdles. Despite the above mentioned limitations, HDC models are valuable tools in mathematical biology for studying complex biological phenomena that involve both discrete entities and continuous fields, such as tumor growth and invasion.

## 4. Conclusions

In this study we investigated the influence of matrix stiffness on the growth and invasion of human glioblastoma tumoroids using a PEGDA-printed tumor-on-a-chip platform. By incorporating varying concentrations of collagenase to create an inhomogeneous collagen matrix, the study successfully demonstrated the strong dependency of tumor behavior on the stiffness of the surrounding extracellular matrix. The results showed that tumoroids exhibited higher growth rates and invasion lengths in response to higher concentrations of collagenase. These findings highlight the potential of investigating the impact of various matrix characteristics on tumor growth and invasion. Furthermore, the study employed a hybrid modeling technique that combined a continuum reaction–diffusion model and a discrete model to accurately capture the growth and invasion in an inhomogeneous environment. The agreement between the experimental results and the model predictions further confirmed the validity and potential of this approach. Extending the tumor-on-a-chip platform to incorporate other components of the tumor microenvironment could offer a more comprehensive representation of the complex tumor–stroma interactions. This advancement would enable the study of how different cellular and extracellular components contribute to glioblastoma progression in response to varying matrix characteristics. Ultimately, such investigations may unveil novel therapeutic targets and strategies for combatting glioblastoma’s aggressive behavior in a more physiologically relevant context.

## Figures and Tables

**Figure 1 biomimetics-08-00421-f001:**
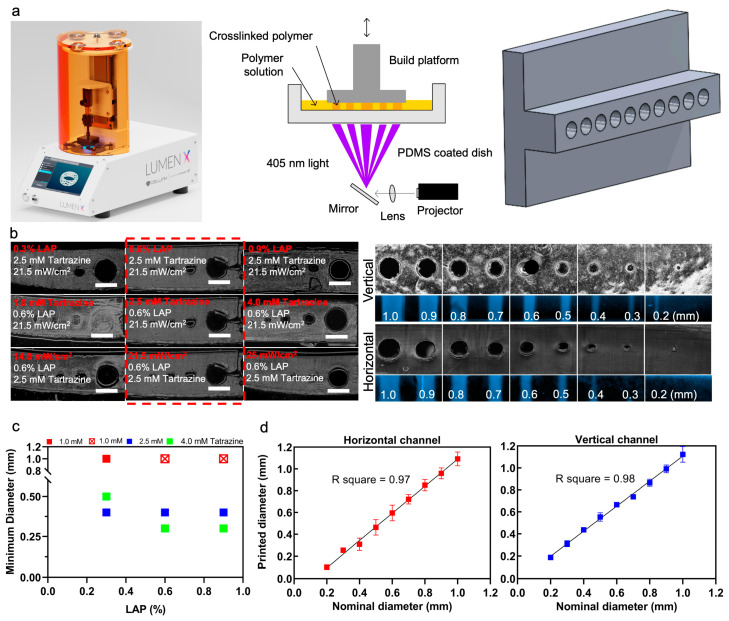
Optimizing printing variables for horizontal and vertical channels. (**a**) Image of the Lumen X DLP printer (left), schematic of basic DLP printer in operation (middle), and an example of a CAD model that was used to print vertically oriented channels of different sizes (right). (**b**) The change in printable channel diameter when a single component is varied in each row, such as LAP concentration, tartrazine concentration, or projector power, from the optimized prepolymer 15% PEGDA solution of 2.04 mM LAP, 2.5 mM tartrazine, 21.5 mW/cm^2^ power intensity, and 5 s/layer exposure time (left). Microscope images were taken of the sliced cross sections of the PEGDA channel constructs which featured channels with diameters of 1.0, 0.50, 0.40, 0.30, 0.20, and 0.10 mm. Channel cross-sections and top view of vertically printed and horizontally printed PEGDA channels (right). Purple fluorescent dye was injected into the channels to confirm that they were hollow. (**c**) Graph demonstrating the effects of tartrazine and LAP concentration in 15% PEGDA, 21.5 mW/cm^2^ power intensity, and 5 s/layer exposure time. When tartrazine concentration was too low, hollow channels could not be printed, as demonstrated in the case of 1 mM tartrazine with 2.04 mM and 3.06 mM LAP (red crosses). (**d**) Ten channel diameters from 1.0 to 0.1 mm were printed in optimal condition and compared with nominal diameters. Scale bars are 1 mm.

**Figure 2 biomimetics-08-00421-f002:**
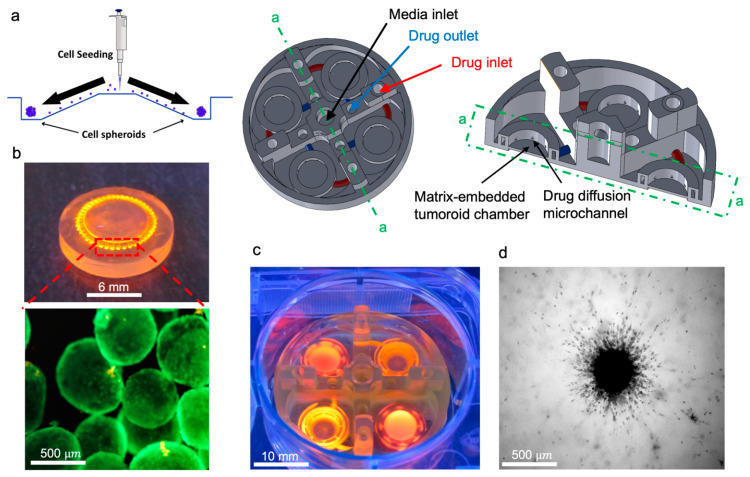
In vitro tumoroid invasion platform. (**a**) Single cell suspension seeded through the loading zone of a self-filling microwell array. (**b**) Tumoroids were formed after four days of culture and were transferred into the tumor-on-a-chip platform. (**c**) The platform was capable of growing tumoroids in four different chambers, each addressed separately, with an inlet and outlet for collagenase treatment. (**d**) Tumoroids embedded in bovine fibril collagen hydrogel were loaded into the open surface tumoroid-on-a-chip platform and their growth and invasion were monitored over time.

**Figure 3 biomimetics-08-00421-f003:**
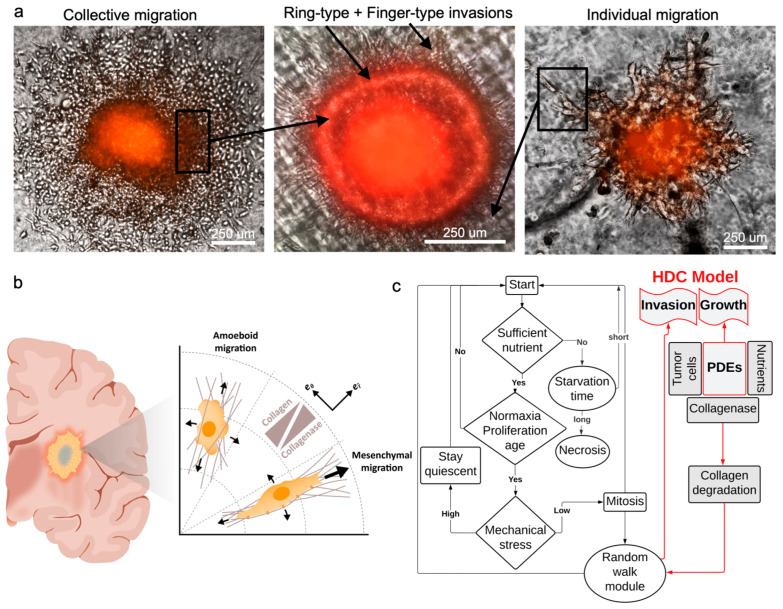
Patterns of hGB invasion. (**a**) Individual and collective migrations contribute to the invasion patterns. Finger-type pattern is mainly derived from individual cells migrating via mesenchymal motion, and ring-type pattern manifests the collective migration mainly via amoeboid motion. (**b**) Mechanism of cellular migration includes directional (mesenchymal) and random (amoeboid) motions, which is captured using a hybrid discrete-continuum model (HDC). (**c**) The model combines modules of cellular processes and random walk with continuum fields of variables, such as cell and nutrient concentrations.

**Figure 4 biomimetics-08-00421-f004:**
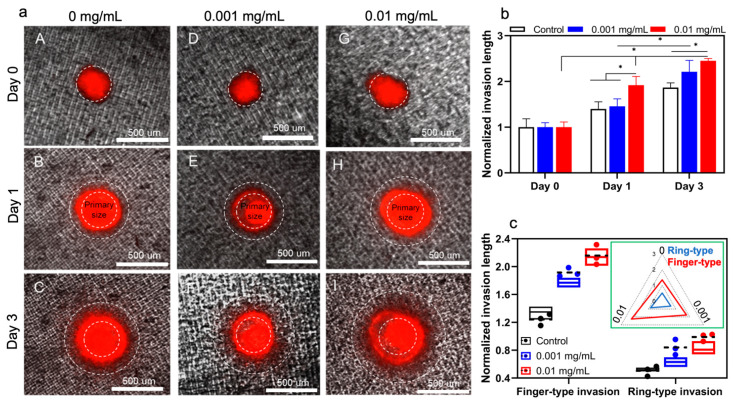
The in vitro invasion of hGB tumoroids. (**a**) Tumoroids exhibit both ring- and finger-type invasion patterns in response to different concentrations of collagenase; 0 mg/mL (A–C), 0.001 mg/mL (D–F), and 0.01 mg/mL (G–I). (**b**,**c**) Effects of collagenase concentration on overall invasion length and invasion pattern are quantified and compared with model predictions (i.e., circles with dashed lines). The inserted figure shows a higher increase in finger-type invasion compared to ring-type invasion length.

**Figure 5 biomimetics-08-00421-f005:**
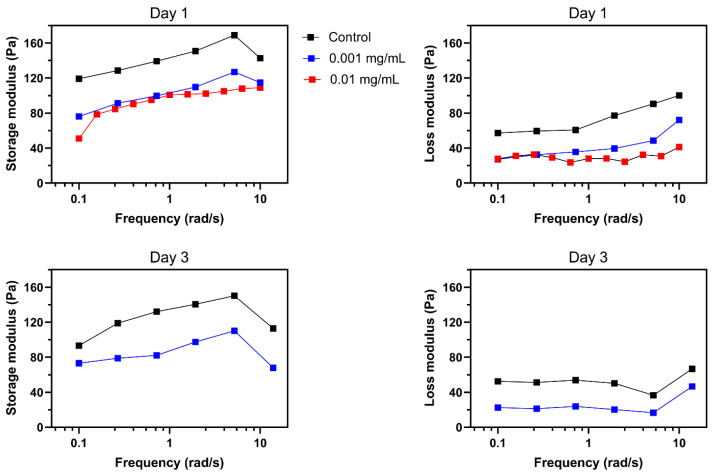
Effect of collagenase on mechanical properties of collagen. A decrease in storage and loss moduli was observed in response to 24 h of 0.001 and 0.01 mg/mL collagenase. Further reductions were observed after 72 h of treatment with 0.001 mg/mL of collagenase. Missing results for 72 h of 0.01 mg/mL collagenase is due to the significant degradation of collagen.

**Table 1 biomimetics-08-00421-t001:** HDC model parameters.

Parameter	Definition	Value	Ref.	Note
*D_i_*	Cell diffusivity	1 × 10^−8^ cm^2^·s^−1^	[47]	
*D_n_*	Nutrient diffusivity	4.2 × 10^−6^ cm^2^·s^−1^	[48]	
*DM*	Collagenase diffusivity	1 × 10^−9^ cm^2^·s^−1^	[49]	
*C* _0_	Initial concentration of cells	1 × 10^4^ mg·mL^−1^	[50]	
*n* _0_	Nutrient supply concentration	4.5 mg·mL^−1^	Gibco DMEM	Glucose is taken as the main component of nutrients
*f* _0_	Initial concentration of matrix fibers	1 × 10^−9^ M	[49]	
*χ_hap_*	Hapto-taxis coefficients	2.6 × 10^3^ cm^2^·s^−1^·M^−1^	[51]	
*P_cr_*	Critical stress	1 kPa	[52]	Maximum allowable stress is 5 kPa
$\eta_{p}$	Rate of cell proliferation	Function is taken form ref.	[39]	
$\lambda_{p}$	Rate of nutrient consumption	Function is taken form ref.	[39]	
ζ	Rate of collagenase binding	1 × 10^−6^ s^−1^·M^−1^	N/A	Value is proposed
*δ*	Rate of collagen degradation	1 × 10^−2^ s^−1^·M^−1^	N/A	Value is proposed

## Data Availability

Data will be made available on request.

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
