# Peer review of "3D-Printed Tumor-on-a-Chip Model for Investigating the Effect of Matrix Stiffness on Glioblastoma Tumor Invasion"

_biomimetics, 2023, doi:10.3390/biomimetics8050421_

Round 1
Reviewer 1 Report
-
The paper should provide a clear and comprehensive explanation for how gaining insights into tumor invasion mechanisms can directly contribute to the enhancement of treatment strategies.
-
The introduction section would benefit from a more streamlined and coherent organization of information.
-
The rationale for selecting matrix stiffness as a focal point for investigating the invasive behavior of tumoroids should be explicitly outlined amidst the range of factors under consideration.
-
The paper lacks an explanation of the statistical analysis methodology employed, necessitating a clarification in this regard.
-
It is recommended to thoroughly compare the impact of matrix stiffness on invasion length and pattern with existing findings in the literature.
-
The paper should expound on the potential alignment between the obtained results and in vivo data, offering insights into how the findings could be validated through real observations.
-
Further elaboration is needed regarding the specific techniques and procedures employed by this platform to identify novel therapeutic targets and formulate strategic approaches, enhancing the reader's understanding of this aspect.
Minor modification is required.
Author Response
We appreciate the reviewer’s careful reading and constructive comments and suggestions. We have taken all comments into account and made revisions, accordingly. Comments are listed below (C) followed by our response (R) in the attached file. Changes to the manuscript are highlighted in yellow.

Reviewer 2 Report
Overall, this article described the process of building an in silico hybrid mathematical model for the prediction of glioblastoma tumor evolution and the investigation of interacting variables. This advancement would enable studying different cellular and extracellular components that contribute to glioblastoma progression. Detailed comments are listed as follows:
1. What’s the major limitation of this so-called hybrid discrete-continuum model? Since this model can cover the process both of micro and macro scales, does it miss any information or emphasis more on certain perspectives?
2. In Figure 1b, some of the sub-figures are repeated and it’s hard to understand the process of optimization and the logic of the parameters changing. Could you please modify this figure and made it clearer? Moreover, please explain this process in the discussion.
3. Please add the legend of the red cross in Figure 1c. Moreover, add the other printing parameters in the figure caption.
4. Is Figure 1d show the results in optimal conditions?
5. Line 157, check the typo of the cell concentration.
6. In Figure 2a, the cell seeding was done manually and flowed into different channels through one inlet. How to make sure the densities of cell spheroids in different channels are close to each other?
7. Please add more discussion in Section 3.2, including the advantages and disadvantages of this technique for modeling this process.
8. As described in the methods section, the cultured U251 tumoroids were added to the chips. Then, how to determine the proliferative cells (Cp)?
9. Have you tested the relationship between the rate of collagen degradation and the concentration of collagenase? Does the assumption in line 290 and line 291 have a range?
10. Any specific reasons for choosing storage modulus and loss modulus as the representative mechanical properties? What’s the implication of tumor invasion? Are these numbers close to the actual ECM in the human brain?
N.A.
Author Response

(The authors gave the same response as above.)

Reviewer 3 Report
The current work by Amereh et al describes the development of an on-chip platform for studying the effect of matrix stiffness on glioblastoma tumoroids (U251) cultured in collagen hydrogels. The study is well hypothesized and the design of experiments for the work is balanced, providing sufficient data to validate their hypothesis.
Here are few suggestions which the authors should consider for improving the manuscript:
1. It would be better if photo-initiator (LAP) and photo-absorber (tartrazine) used be represented in same units (mM or in %), as this would give a clarity at what ratio the print quality is coming out fine.
2. Figure 1b, the scale bar needs to be defined.
3. The rationale for using 15% PEGDA must be mentioned.
4. Instead, of mentioning the power output a 50% (section 3.1, line 228) it must be mentioned in terms of W/cm2 so that the same resin can be tested/ reproduced in another DLP printer (other than Cellink’s LumenX which has been used in the current study).
5. Section 1 (line 77) Secction 2.6, the use of the term 3D bioprinted microfluidic chamber is not appropriate. There are no cells encapsulated in the bioink. It is just a 3D printed chamber to which collagen gel is added. The authors need to clarify this.
6. Section 3.2, the authors need to provide more clarity on the STL design files. They claim it to be microfluidic device, but where the cultures done in static condition inside a 12 well plate (Figure 2C)? Was the entire chamber immersed in culture media? How does the collagenase concentration maintain in that case?
7. HDC model has been investigated in this study to evolution of tumour with respect to matrix stiffness. However, the authors need to add a note on how matrix stiffness is crucial in GBM and the implications of it during tumour formation.
8. Section 3.3, line 284 “In this model, state variables such as concentrations of cells, nutrients, collagen, and collagenase were assumed to be continuous”, does it reciprocate the culture conditions carried out? Authors must add more explanation to justify this.
9. Authors must add a translational standpoint of the 3D printed chamber they have developed and for the on-chip application, of how it can be used. The discussion section was weak and it needs to be improved.

Some minor typesetting and typographical errors need to be corrected.
Author Response

(The authors gave the same response as above.)

Round 2
Reviewer 2 Report
Good job! The revised manuscript has seen a significant improvement in quality, and it's evident that the authors have adeptly addressed all my concerns.
Cheers to your hard work and dedication!
No further comments.